# Contemporary European Welfare State Transformations and the Risk of Erosion of Social Rights: A Normative Analysis of the Social Investment Approach

**Gianluca Busilacchi** [1,2,*] and **Benedetta Giovanola** [3,4]

1   Department of Economics and Law, University of Macerata, 62100 Macerata, MC, Italy
2   Minda de Gunzburg Center for European Studies, Harvard University Cambridge, MA 02138, USA
3   Department of Political Science, Communication and International Relations, University of Macerata, 62100 Macerata, MC, Italy
4   Department of Philosophy, Tufts University, Medford, MA 02155, USA
*   Correspondence: gianluca.busilacchi@unimc.it

**Abstract:** Over the last decades, there has been a huge debate on the transformations of the European Welfare State. The issue of its financial sustainability together with the emergence of new social risks has put under pressure the traditional model of social protection and created the conditions for a change in the gist of the welfare state provisions. In this context, the social investment approach has become an emerging reference paradigm to tackle new social risks and meet the need to recalibrate the European welfare state and ensure its economic sustainability. However, despite this success, social investment still seems to be a rather ambiguous concept, too vague to result in precise and univocal policy prescriptions and open to the risk of a stretch of its interpretation by neoliberal politics, to erode social rights. In this paper we propose a theoretical framework to better clarify the normative ground, the moral foundation and political justification of the social investment approach and to understand whether it can avoid the risk of the erosion of social rights.

**Keywords:** social investment; Welfare State; social rights; normative framework; responsibility; capability; distributive justice; luck egalitarianism

## 1. Introduction

Over the last years the social investment approach (SIA) has become an emerging reference paradigm to tackle new social risks and meet the need to recalibrate European welfare states and ensure their economic sustainability [1–3].

The core of the SIA is that social policies should promote individual empowerment, increase worker opportunities and employability, decrease welfare dependency and sustain economic growth. The underlying idea is that welfare state intervention—understood as a regulation of resource allocation—should not depend merely on a principle of justice in distribution, aimed at repairing and compensating market failures and ensuring social rights for every citizen, but should rather be conceived of as a productive factor to boost opportunities of development, nudging the individual towards independence from the welfare state.

This principle has proven to be successful both at the institutional and the academic level, since it has met political urgencies and scientific mainstream culture: the necessity of recalibrating the social protection structure and expenditure on the one side, and the shift of the European welfare state towards individual responsibilization [4,5] on the other side.

With the adoption of the Social Investment Package in 2013 by the European Commission [6], the SIA has been assumed by the EU as a reference paradigm[1] to steer the European social model towards the goal of combining economic growth and social inclusion, culminating more recently in the adoption and implementation of the European

Pillar of Social Rights [8]. The interpretation of the SIA as a paradigm in the above sense can be fruitful from a political perspective, especially to spread the idea even in different political contexts; however, it is problematic on the scientific side, where clear concepts and definitions are the condition for researching purposes.

In fact, despite its success, social investment (SI) itself seems to be a rather ambiguous concept (a "quasi-concept", as some have argued), too vague to result in precise and univocal policy prescriptions. Moreover, besides its potentialities, the SIA has also been exposed to several critiques, concerning its empirical use, originated by conceptual and methodological flaws.

It has been argued that the SIA presupposes a context of economic growth to produce an appropriate job demand [9]; that it presupposes that the old risk structure has been substituted by a new one and that passive policies are evaluated in negative terms, while active ones in a positive way [10]; that there is a weak causality of the 'investment return' of the policies in economic terms, that is, it is very hard to estimate the SI policies' actual direct effects on growth and employment; that it is not so clear what kind of 'return of investment' is expected [11]; that the SI concept is too vague and addresses too different policy instruments; that it creates a Matthew effect [12]; and that it ignores asymmetries in the labour market by considering all individuals the same way and ignoring pluralism [2].

Most of the critiques that emerged during the first wave of research on social investment were based on a juxtaposition of social investment policies and social protection policies, and recently found an answer through a better classification of the social investment policies and their specific roles especially [3,13,14]. The most important critique for the purpose of this paper is that the SIA is an incomplete paradigm because there are different understandings of SI that have different normative underpinnings regarding social citizenship and social progress [15]; in particular, it seems crucial to solve the issue of whether the ultimate goal of the SIA is the sustainability of the welfare state through progressive de-responsibilitization of the State, or the empowerment of individual capabilities. In fact, the SIA can be interpreted in a positive manner, as a tool to widen the range of citizens' social rights and increase their empowerment and capabilities, or in a negative and punishing sense, as a way to restrict social rights to deserving recipients.

The thesis of paper is that in order to identify the ultimate goal of the SIA and to solve the conceptual and methodological problems it raises, it is necessary to analyse and clarify its normative background [15]. Our main hypothesis is that in order to clarify the normative framework of the SIA, it is necessary to focus on the relationship between social risks and social risk protection and to uncover the latter's moral foundation and political justification. As we will show, the inquiry into the moral foundation of social risk protection allows us to identify the ethical principles that would make the distribution of social provision fair according to the SIA; these principles in turn provide the normative ground for justifying (or not) public intervention, through welfare state policies aimed at protecting social rights.

Here we understand "social risks" as the probabilistic phenomena that hamper the satisfaction of one's needs, through a deprivation of primary goods, and "social rights" as the entitlements to enjoy access to primary goods. Solving this Gordian knot will help us answer crucial questions such as whether the SIA supports or opposes social protection systems, whether it is an alternative or a complement to the traditional welfare state [10], based on social rights achievement. It will also help solve other significant problems, such as whether a welfare state reform based on the SIA might substitute social insurance and social assistance schemes, or might be integrated into them and how and, more importantly, whether the SIA can support the achievement of social rights or rather erode them.

In order to develop our thesis, we first propose a general normative framework for analysing social protection systems and show how it helps understand both the traditional welfare state mechanisms and the emergence of the SIA (Section 2); second, we highlight the main reasons that can explain the emergence and success of the SIA, as well as some of its shortcomings (Section 3); third, we uncover both the moral foundation of the social protection system based on the SIA—claiming that it is based on individual choice

and responsibility—and its political justification, showing that it can be framed in luck-egalitarian terms: here we also highlight that this view entails some potential problems and shortcomings for a fair distribution of social rights (Section 4); finally we conclude by trying to answer whether the SIA is encouraging or eroding the achievement of social rights (Section 5).

## 2. The Normative Framework of Social Risk Protection: Responsibilities and Capabilities

Historically, the welfare state has represented one of the main regulation mechanisms of the distributive effects of the market, namely the protection from social risks for individuals in need. Since its creation, in the 19th century, social insurance has protected workers' economic conditions when workers are without a job and consequently without a salary, and thus exposed to social risks.

At a general level, addressing the issue of the normative framework of the welfare state implies focusing on the way in which the relationship between social demand and social supply is considered and valued. More specifically, on the social demand side, it is necessary not only to understand how the social risks embedded in a society can result into a collective demand, but also to pursue a micro analysis of individuals' expectations, ideas of well-being, goals in life, etc., that is, what people have reason to value [16]. This is important for two reasons: first, social risks do not affect all individuals, who are exposed to different levels of risk according to their position in the social structure, in the same way [17]. As to their individual conditions, we can say therefore that there is a pluralism of social risk exposure. Second, the social demand does not depend only on the social risks that can occur, but also on the evaluation of losses of wellbeing in the individual's perception. Moreover, on the supply side, it is necessary to take into account the risks that are considered relevant (to be protected) by public policy and by governments, but also the goals of a society in general, i.e., the outcomes in terms of progress that are considered socially valuable; as sketched below, this recalls a moral foundation and political justification of the welfare state's final goal (see Section 4).

As we can see in Figure 1, the mechanism of social risk protection involves 'community relations', in particular the relationship between the state and individual potential recipients of a social policy. The intervention of the state through a public action, by taking public responsibility in the social protection mechanism, is legitimated by the fact that some social risks are considered as deserving public protection by the community, for reasons of reciprocity [18] and a combination of reciprocity and solidarity [19], and therefore to carry a part of the individual responsibility of protection at public level.

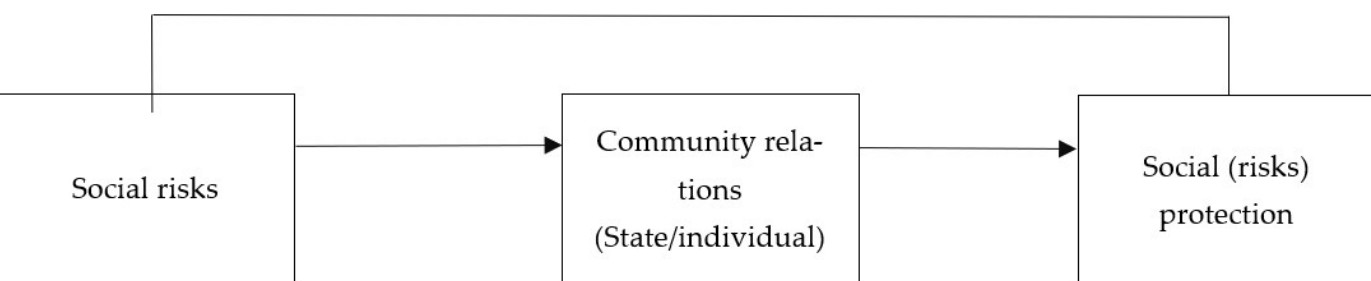

**Figure 1.** The social protection mechanism: the conversion process.

The protection mechanism from social risks is indeed always a mix of a public intervention and individual effort: the different mix of these two elements depends basically on the collective assumption of the charge of social protection. The different level of conditionality can represent from this point of view the reciprocity request for individuals to be part of a community, to be a citizen; even when the welfare state ensures social rights to citizens, this notion of citizenship passes through some required minimum behaviour (paying taxes, etc.). In different social protection mechanisms, as we will see, this notion

can be interpreted in different ways; using this perspective, it is possible to argue that a social policy paradigm shift occurs when the degree of reciprocity, namely the nature of the 'community relations', changes.

The responsibility taken—both at an individual and public level—represents the willingness to contrast social risk, not to ensure the final result, that depends on the (mixed, public and private) capability of undertaking it.

The public capability to contrast social risks is related to the financial sustainability of the welfare states and also to the effectiveness of public policy implementation; individual capabilities are very different among individuals, and they represent the plurality of social risk perceptions and exposure and the different level of agency in contrasting them.

The two aspects—responsibilities and capabilities—are therefore very connected to each other: it is a public responsibility of the State to ensure that all citizens are capable of real agency [5] when they are asked to be responsible for coping with social risks. To sum up, the micro–macro connection of the 'conversion process' showed in Figure 1 is important to consider if we want to investigate the normative welfare state framework, focusing on two different—though connected—aspects, which are both crucial: the first aspect is the individual–State relationship, that is, the 'community relations' that legitimize public policies; the second is the way the sum of individual levels of wellbeing becomes a collective welfare function. The latter aspect addresses the question of whether the collective welfare of a society is made merely of the sum of individual levels of wellbeing, or if it is more than this sum; we can assume that it is more, because there are social assets, such as social cohesion and collective trust, that go beyond individual wellbeing (even if they influence it).

The former aspect concerns the legitimation of social policies and claims that it rests upon the social contract between individuals and the state. Here, a specific theory of the state is applied to welfare state analysis: social protection would "hence correspond to a form of the State" (see [10], p. 55) because it is a collective institution, a system that combines macro relationships and "relation of individual citizens to the state, the economy and the family" (see [20], p. 419). In the framework that we assume here, it is the relationship between responsibilities and capabilities at individual and public level that determines the kind of community relations between individual and the state.

If we look at the history of the European welfare state through the perspective of community relations, it is possible to highlight the nature of the changes in the social protection mechanism, and therefore to assess whether a paradigm shift has occurred.

As shown in Table 1, it is possible to apply this normative framework to analyse the traditional forms of social protection.

**Table 1.** Social risks and protection mechanisms in different ages of the welfare state.

| Age of the Welfare State | Social Risks | Community Relations | Social Protection |
|---|---|---|---|
| Birth of modern welfare state | Actuarial and circumstantial risks | Contributive-reciprocity relations between workers and State | Social insurance |
| Golden age of the welfare state | Social class risks | Citizen solidarity as foundation of the community | Social assistance and social security in addition to social insurance |
| Neoliberal age | Individual risks | Social contract between deserving individuals and the state | Welfare state retrenchment |
| Social investment welfare state age | Neither actuarial, nor predictable risks (plus old ones?) | Activation as reciprocity, or activation for new social rights? | Social investment as workfare or as human development? |

When the first forms of social protection arose at the end of the 19th century, they aimed to face "actuarial and circumstantial risks" [21], i.e., the occasions on which workers had to cope with circumstances that prevented them from having a salary were rare and predictable. Bismarckian social insurance was the right social protection mechanism to tackle these kinds of risk; it was the first time that the state took on the responsibility of protecting the social risk of impoverishment for a specific category of people, that is, the workers. The nature of the social insurance as a protection mechanism is then based on very clear 'community relations', based on a reciprocity relation between workers and state, automatically ensured by the contributory mechanism.

When, after the Beveridge reform, other social protection mechanisms, namely social assistance and social security, were added to social insurance schemes, the welfare state began to represent a fundamental aspect of the European social model. During the so-called golden age of the welfare state this model guaranteed wellbeing to European citizens, limited inequality and strengthened social cohesion in the old continent. The community relations at the base of the social protection mechanism were broadened, compared to the former era, including all citizens, because the principle of mechanic reciprocity was abandoned in favour of an inclusive solidarity as the foundation of citizenship. Consequently, the increase in social expenditure was the effect of a wider conception of social risks deserving of the responsibility and protection of the state.

Therefore, the welfare state combined two important institutional aspects: the regulation of market failures, to protect citizens against social risks, and the foundation of social rights of citizenship. We contend that the relationship between social risks and the protection mechanism has represented the core of the normative framework of the modern welfare state since the Beveridge reform.

Only when, after the Beveridge reform, did the social risks of poverty, ignorance, disease and old age cease to be linked to the specific condition of providing social contributions, and started to be considered worthy of protection by and for the whole community through general fiscality—with the establishment of social assistance and social security pillars—, the social protection system started to build the substantial basis for citizens' statutory rights.

However, the normative framework of the so-called Fordist model of the welfare state, which constitutes the basis of the European social model, combining economic development and social cohesion, entered into crisis in the last decades of the 20th century [22]. The retrenchment of public responsibility in taking on social risk protection led to a retrenchment of social expenditure and in general to a shift toward individual responsibility in taking care of oneself. Community relations in this age have the form of a social contract, in which the political justification for welfare state intervention relies on strict reciprocity with the social policy recipients; the individual has to be deserving of collective protection and the enforceability of social rights is then conditioned to specific duties.

The question at present is whether the new, emerging model of the welfare state has already found a paradigm of reference that has a sound structure from the normative point of view. This is crucial not only for analytical reasons, but also because the recalibration of social protection systems and the way in which welfare states are facing social risks and enabling social rights are the basis for the future of the European citizenship.

Even if the social investment state has been proposed as a new paradigm, at least two main doubts arise, following the analytical framework presented here. First, it is not clear whether the social investment state must tackle only the new social risk structure or also the traditional risks, for instance, the ones related to impoverishment. Second and more importantly, the social protection mechanism put into play by the social investment and the community relations at its core are still ambiguous and open to a double interpretation.

On the one hand, the activation of individuals for tackling new social risks can be interpreted in a 'positive' way, as an empowerment of their capabilities and enrichment on the side of their social rights. In this case, the recalibration of the welfare state could be

seen as the adequate way of taking on a new public responsibility and founding innovative community relations.

On the other hand, the same social investment policies can be interpreted as a shift from a public responsibility toward individuals, more or less in line with the neoliberal retrenchment of the welfare state and its strict reciprocity principle. Under this interpretation, the activation of social policy recipients might represent an increased conditionality over the welfare state, to ensure social rights only to deserving individuals who accomplish specific workfare duties. In this case, there would be an individual capability-set contraction and the risk of an erosion of both social rights and the solidarity mechanism.

In conclusion, a possible double interpretation of the welfare state of activation and social investment seems to emerge, and this makes it necessary to clarify the normative framework of the contemporary social protection system. To make this clarification, first we will analyse the contemporary challenges to the European welfare state and the emergence of the SIA and then focus on the moral foundation and political justification of this social protection system.

## 3. Contemporary Challenges to the European Welfare State and Emergence of the Social Investment Approach

Over the last decades there has been a huge debate on the transformations of the European welfare states (among others: [1,2,22,23]). These transformations have been connected to the crisis of the Fordist social model and its system of protection from social risks. There are several reasons for this crisis, which opens a new, post-Fordist era of the welfare state, also called the "silver age of the welfare state" [24]. Despite differences among states, some common trends and factors of change can nevertheless be identified as typical of the evolution of all the European welfare states through the last decades.

First, all European countries have been facing the issue of the financial sustainability of the welfare state; the increase in the dependency ratios—also because of the aging population—and in social expenditure determines budget constraints, especially in a period of globalized and competitive economies. In this regard, a "permanent austerity of the welfare state" [25] has arisen, with the effect of becoming the "politics of welfare state retrenchment" [26]; the effect of this first trend has been the decreasing capability of the state to protect social risks through public policies.

Second, several social and economic changes, such as the globalized economy, new flexible labour markets, increasing female participation in work, and new demographic trends determine the emergence of new social risks [23]—such as the precariousness of work, the long-term care of non-self-sufficient people and the reconciliation of work and private life, especially for women—that have increased individual needs. These social risks are new from the point of view of their structure; they are less predictable than in the past, less related to social class and work activities, less connected to a single period of life. Furthermore, most new social risks present differences in terms of the nature of the protection required, being linked with the process of "individualization" [27]. The latter also determines the problem of the representativeness of social risk protection, which changes the political framework of the welfare state; the possible effect of this second trend is therefore to weaken the legitimization of welfare state intervention and to transform the relationship between individual and collective responsibility in the protection from the new social risks. To answer these new social demands, it is necessary either to empower individual capabilities [16] or to increase the social protection system offered by the welfare state.

Third, the widespread of neoliberalism has fostered a change in the relationship between the individual and society in general [28], with a specific impact on the role of responsibility in the protection of social risks [5], shifting the role of this protection from the state to the individual. In this regard, the effect has been the transformation of the relationship between individual and public responsibility within the European welfare states.

To sum up, in the current era all these common trends have reduced public capabilities for social risk protection, increased individual responsibilities and, more generally, weakened the legitimization of public intervention in favour of individual intervention in the social protection domain.

The welfare state's attempt at recalibration to face this new risk structure produced a shift, especially towards a new 'activation policy' approach; whilst in the past, the way to ensure social rights was based on a one-way provision of primary goods from the state to the recipients, at present, social inclusion is intended as a process that starts from the opportunity to access primary goods that is not guaranteed anymore per se, but requires an individual's commitment.

The main hypothesis underlying this change is related to the fact that the 'old risk' structure required a compensatory solution by social policy, whereas the 'new risks', because of their specific features, need the recipients' activation and responsibility towards social inclusion and work participation.

The post-Fordist model of the welfare state in Europe, in all its different versions, from 'workfare' and 'activation' to 'social investment' (among others: [13,14,29]) in this perspective can be seen as a change in the balancing of the private/public responsibilities and capabilities in the protection against social risks from the past.

The change occurred not only in the design of social policy but more deeply in the normative background of the welfare state, especially in the relationship between the state and individuals, namely in the social contract. If, in the traditional social protection mechanism, the relationship was between a state as the "giver" and an individual as a "receiver", now the recipient of the policy is asked also to be a "doer" [30]. There are two different perspectives in which the principle of activation can be interpreted [31,32]: one is the 'enabling' perspective, through which the recipient of social policy can empower his or her capabilities (in Sen's words) to increase his or her opportunities in life; the other is the 'binding' perspective, with the aim of increasing individual responsibility for social risk protection, through work participation and decreasing welfare dependency. In the former perspective, the social rights of individuals do not decrease; rather, they increase: to face the new social risks the individual has the right to be protected both in the traditional way and also in order to increase their capabilities for social inclusion in the 'new' society. From the latter perspective, a strict interpretation of the concept of responsibility seems to drive the policy approach; the social rights perspective is consequently reduced and the 'receiving' side is conditioned upon whether and how much the 'doer' can give back to the community. In the harshest interpretation, social rights do not exist anymore by themselves as statutory rights, but only insofar they have corresponding duties. The path taken in one way or another by the European welfare states during the last decades depends on welfare regimes and policy directions chosen by the different governments, from a rather 'capability-oriented' approach in the Nordic countries to a strict workfare approach in the UK. It is in this context that the SIA arose and started to represent a variant of the 'active welfare state' and 'enabling state' perspective, centred on the collective return that an investment in human capital might generate. The core of the SIA is indeed considering the welfare state not mainly as a regulation mechanism of the distributive effects of the market (as was the case in the Bismarck and Beveridge eras), but as an investment in human capital that can produce positive 'returns' for the collectivity in terms of increasing individual responsibility for the social risk protection through participation in the labour market and growing the economy (and reducing welfare dependency). SIA is undoubtedly a promising paradigm to tackle new social risks and meet the need to recalibrate welfare states and ensure their economic sustainability. On the one hand, it can represent a new political direction assumed by the EU to modernize its social model and to adapt it to the new economic context; on the other hand, it can represent an answer to the challenge of creating a post-Fordist welfare state. However, besides its potentialities, the SIA has also been exposed to several critiques; among them, as already mentioned in the Introduction, the most significant for the purpose of this paper is that the SIA is an incomplete paradigm

because its normative framework is unclear. The effect of the unclear normative framework of the SIA is particularly significant on the policy-prescription side; as noted by Barbier [10], we can understand the SIA in two different ways, "with" or "against" traditional social protection. This means that by interpreting the SIA in a narrow sense, SI policies face only new risks, considering the old risks and the "social rights perspective" as outmoded and placing attention on the increase in individual responsibilities for the social risk protection mainly through a work-activation perspective. In a broader definition, the SIA integrates traditional social protection and the policy instruments for new risks with traditional assistance against old risks (such as minimum income policies or social insurance); social rights may then be understood in a more capability-oriented perspective and the attention is on individual empowerment and capabilities in facing social risks.

Recently, Hemerijck has supported the latter interpretation, arguing that SI policies are a mix of interventions within which there is room also for "buffer" policies, namely minimum income policies to ensure basic social rights for all (see [13,14]). The problem is that the active SIA has also influenced the contemporary minimum income policies in all Europe, increasing their conditionality and restricting them to deserving, active recipients. Therefore, the SIA, without a solid guarantee of basic social rights protection, seems to recall the importance of individual choice and responsibility in contrasting social risks, rather than the empowerment of one's capabilities. To better develop this point, it is important to analyse the moral foundation and political justification of the SIA.

## 4. The Moral Foundation and Political Justification of the Social Investment Approach

Addressing the issue of the normative framework of the welfare state means dealing with its moral foundation and political justification [33]. In this section, we deepen our analysis of the normative framework underlying the SIA and uncover its moral foundation and political justification; in particular, we claim that there are reasons to interpret the SIA in the narrow sense specified above (Section 3), as its moral foundation is ultimately based on individual choice and responsibility and its political justification can be traced back to a luck-egalitarianism approach, that is, an approach to distribution which is based on individual choice and responsibility.

Luck egalitarianism is a major approach to distributive justice, whose core claim is that a distribution, in order to be fair, ought to be 'ambition-sensitive' (or choice- or responsibility-sensitive) and 'endowment-insensitive'. The former condition requires that distribution depend on choices made by individuals in ways that reflect their option luck, whereas the latter condition requires that distribution does not depend on differential brute luck. Brute luck is fortune, over which individuals have no control, while option luck is the upshot of risks that were, in some sense, deliberately taken and for which individuals have responsibility. To put it in Dworkin's words: "Option luck is a matter of how deliberate and calculated gambles turn out—whether someone gains or loses through accepting an isolated risk he or she should have anticipated and might have declined. Brute luck is a matter of how risks fall out that are not in that sense deliberate gambles. If I buy a stock on the exchange that rises, then my option luck is good. If I am hit by a falling meteorite whose course could not have been predicted, then my bad luck is brute (even though I could have moved just before it struck if I had any reason to know where it would strike)" (see [34], p. 293).

In other words, if a person ends up badly and another one well due to their choices, there is no need for compensation for the former person out of principles of distributive justice, as no one deserves compensation for the bad outcomes of their choices. To sum up: everyone should be held responsible for the choices they make.

The luck-egalitarian approach has first been put forward by Ronald Dworkin in his influential article "What is equality?" [34] and was born as an attempt to recover the importance of individual choice and responsibility within the egalitarian framework (see [35]), as well as to rebut the libertarian critique [36], according to which justice would not require any (re)distribution but would just be a matter of individual choice and responsibility,

regardless of any eventual resulting inequality; in fact, (re)distribution, in libertarian approaches, is considered as deeply unjust, as it would be a sort of "slavery" of the talented, who would be deprived from the fruits of their talents and their labour 'just' to benefit others who do not deserve the enjoyment of those fruits.

In their attempt to recover the importance of individual choice and responsibility within the egalitarian framework, luck egalitarians draw insights and at the same time go beyond John Rawls's theory of justice [37], which can be considered the cornerstone of the contemporary debate on theories of justice, mainly framed in liberal-egalitarian terms (see [38]).

Rawls, as is well-known, proposes a twofold principle for arranging socio-economic inequalities: the principle of fair equality of opportunity and the difference principle[2]. The principle of fair equality of opportunity is of particular interest for our discussion here; it entails that that the distribution of shares—and, more specifically, access to opportunities—in any society be not influenced by socio-economic contingencies, that is, by people's different "starting places". Should this happen, the resulting inequalities would be deeply unjust, as they could not be justified by an appeal to the notion of moral merit or desert ([37], p. 7). The principle of fair equality of opportunity not only requires a formal equality of opportunity, ensured, for example, through the legal system or expressed by reference to the idea of "careers open to talents" ([37], p. 57); it entails the promotion of real and substantial opportunities for every person to effectively access the positions they are willing to access and are talented for. To put it in Rawls's words: "The thought here is that positions are to be not only open in a formal sense, but all should have a fair chance to attain them"; in other words, "those who are at the same level of talent and ability, and have the same willingness to use them, should have the same prospects of success regardless of their initial place in the social system. [ . . . ] The expectations of those with the same abilities and aspirations should not be affected by their social class" ([37], p. 63).

The Rawlsian principle of fair equality of opportunity can be read as including a reference to the importance of individual responsibility; the underlying idea, in fact, is that each person exercises their responsibility in choosing their own ends, provided that every choice is made within the range of possibilities determined by the context in which they live (see [39]). Responsibility then, in Rawls's view, is individual, but cannot be fully detached or disembedded from the social context, because social inequality and disadvantages bear on the very possibility and assessment of individual responsibility. Accordingly, there is a sort of "division of labour" between the individual and the society: individuals are responsible for their choices and for the definition of their ends, but at the same time the (basic structure of a) society must secure basic freedoms, fair equality of opportunities and a fair distribution of primary goods to all its members [39].

Differently from Rawls, luck egalitarians emphasize the role of individual responsibility without giving enough importance to its social dimension. Individual responsibility then is considered as the main principle for any fair distribution and leads one to justify a high degree of conditionality in the distribution itself. In fact, the distinction between option luck and brute luck, as well as the importance of individual choice and responsibility as the core 'condition' for a fair distribution, first introduced by Dworkin, still hold in the different variants of the luck-egalitarian perspective (see [40–43]), despite relevant differences among them in other aspects [38].

The overemphasis on individual responsibility has prompted different critiques of luck-egalitarian accounts. A major critique is that of being a too harsh approach in as far as it would eventually lead one to justify the abandonment of the so-called "negligent victims", that is, those who ended up in a bad situation due to risks they have deliberately taken. This abandonment, it has been claimed, would be deeply disrespectful of persons and strictly inegalitarian (see [44–47]), as it would ultimately clash with the acknowledgement of every person's moral equality. Moreover, it would not only have a discriminating impact at the distributive level towards the "negligent victims", but it would also bring about

socio-relational inequalities, as the "negligent victims" would be looked at as less valuable citizens, not deserving equal respect due to the choices they made.

The luck-egalitarian approach helps understand the normative core of the SIA and the importance it attributes to individual responsibility as the condition to access social risk protection and to distribute social rights fairly, without guaranteeing any social right to the "negligent victims". Such an emphasis on individual responsibility, however, has serious shortcomings, especially in as far as it does not acknowledge the importance of starting social inequality and disadvantages and the way they bear on assessments of individual responsibility itself. Unpacking the luck-egalitarian foundation of the SIA helps uncover a potential paradox of the latter, that is, that it overlooks the importance of the social determinants of individual responsibility, and understands social risk protection as conditional upon individual responsibility, thus legitimizing what it should counter. In other words, overlooking the impact of social inequality and disadvantages on the very possibility of exercising individual responsibility, the SIA understands social risk protection in a very weak and partial way, as just a matter of (conditional) distribution ex post, without considering the importance of the enjoyment of social rights ex ante. Moreover, in doing so, the SIA runs the risk of discriminating between "negligent victims", who would not deserve a full enjoyment of social rights, and "deserving citizens", thus eventually leading to inegalitarian outcomes in the access to social risk protection and to an unfair distribution of social rights.

## 5. Concluding Remarks

The transformations of the European welfare state over the last decades are very evident and have been outlined by several scholars. The issue of the financial sustainability of the welfare state, together with the emergence of new social risks, has put under pressure the traditional model of social protection and created the conditions for a change in the gist of welfare state provisions.

In this context, the SIA emerged as a new reference paradigm for steering the contemporary European welfare state and guaranteeing adequate social protection. The core of the SIA—as we have shown—is moving from a traditional recovery vision of welfare state, aimed at decreasing the inequalities generated by market failure through the distribution of social provision, toward an enabling vision of the social protection mechanism, aimed at increasing the empowerment and capabilities of recipients in the contemporary active society. However, we have also shown that the SIA entails some ambiguities: even though it is very fruitful to understand the actual trends of the European welfare state, some of its underlying ambiguities might lead one to interpret it in neoliberal terms and to eventually justify an erosion of social rights. In order to overcome these ambiguities we proposed a normative framework that helps deepen the analysis of the 'social contract' between the state and social policy recipients to better clarify the interactions that the SIA policies might generate and to evaluate to what extent basic social rights are protected. In particular, we claimed that the SIA can be understood as either oriented by principles of individual choice and responsibility in contrasting social risks, or by an empowerment of one's capabilities.

Going a step forward in our analysis of the normative framework of SIA, we delved into its moral foundation and political justification, and uncovered a luck-egalitarian underpinning of the SIA, according to which any distribution (of social rights) ought to be based on assessments of individual choices and responsibilities. As a consequence, the SIA would eventually lead to a conditional distribution if not an erosion of social rights, rather than an empowerment of one's capabilities.

Therefore, without a guarantee of a basic and universalistic social rights foundation, aimed at ensuring a minimum protection for all and at avoiding the punishment of the "negligent victims" and undeserving recipients, the SIA seems to be oriented more on the side of individual choice and responsibility rather than on the capabilities side. This would imply that the SIA would eventually lead to the erosion of social rights. On the contrary, the SIA could turn into a way of empowering individuals only if a preliminary clarification

is carried out: that only by ensuring basic social rights for everyone it is possible to build up social investment policies. In the latter case, the justification of the state intervention on the initial unequal allocation through a regulative mechanism would rely upon a more complex notion of responsibility that is compatible with the development of everyone's capabilities [16].

**Author Contributions:** Conceptualization, G.B. and B.G.; methodology, G.B. and B.G.; software, G.B. and B.G.; validation, G.B. and B.G.; formal analysis, G.B. and B.G.; investigation, G.B. and B.G.; resources, G.B. and B.G.; data curation, G.B. and B.G.; writing—original draft preparation, G.B. and B.G.; writing—review and editing, G.B. and B.G.; visualization, G.B. and B.G.; supervision, G.B. and B.G.; project administration, G.B. and B.G.; funding acquisition, G.B. and B.G. All authors have read and agreed to the published version of the manuscript.

**Funding:** This research received no external funding.

**Institutional Review Board Statement:** Not applicable.

**Informed Consent Statement:** Not applicable.

**Data Availability Statement:** Data sharing is not applicable to this article as no datasets were generated or analyzed during the current study.

**Conflicts of Interest:** The authors declare no conflict of interest.

## Notes

[1]  'Paradigm' here "refers both to cognitive understanding of causal relations between policy efforts and outcomes and to political mobilization behind policy priorities" (see [7], p. 323).

[2]  The difference principle requires that—once the principle of fair equality of opportunity is guaranteed—the overall scheme of cooperation and distribution maximizes the expectations of the worst-off (see [37]).

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
