# Peer review of "Contemporary European Welfare State Transformations and the Risk of Erosion of Social Rights: A Normative Analysis of the Social Investment Approach"

_societies, doi:10.3390/soc13020021_

Round 1

Reviewer 1 Report

The article discusses a very interesting issue. Its structure is very clear and rigorous, and the main thesis is original. Finally, the references are adequate. I don't to see any reason not to accept the article as it is. 

Author Response

Dear editor,

Dear reviewers,

We would like to thank you for your valuable and useful comments to our concept paper “Contemporary European Welfare State Transformations and The Risk of Erosion of Social Rights: A Normative Analysis of the Social Investment Approach”.

The comments were very useful and helped us a lot to clarify our thoughts.

Thank you so much for your consideration of the revised version of our manuscript.

Sincerely,

Gianluca Busilacchi

Benedetta Giovanola

Reviewer 2 Report

Line 85: Please, define ‘social risks’ and ‘social right’ when mentioned for the first time

Line 104: Please, tell something more about the connection between solidarity and reciprocity. Why solidarity? Where from such a notion?

Lines 246-247: maybe a few words can be added to expand the idea that new risks need the recipients’ activation and responsibility towards social inclusion… The idea is crucial and already stated, but maybe it can be further explained.

Lined 314: Ronald Dworkin, not Donald.

§ 4: the title and somewhere in the first lines speak about ‘moral foundation and political justification’, but the section does not mention the differences (if any) between the two. The recalling of lucky egalitarianism is surely the path to moral justification: why speak of ‘foundation’ and how to distinguish moral from political justification?

Author Response

Dear editor,

Dear reviewers,

We would like to thank you for your valuable and useful comments to our concept paper “Contemporary European Welfare State Transformations and The Risk of Erosion of Social Rights: A Normative Analysis of the Social Investment Approach”.

Here below we sum up the way(s) in which we tried to respond to the comments made by Reviewer 2, as Reviewer 1 and Reviewer 3 did not ask for revisions:

  1. Line 85: Please, define ‘social risks’ and ‘social right’ when mentioned for the first time: In response to this comment we defined both concepts (lines 72-73)
  2. Line 104: Please, tell something more about the connection between solidarity and reciprocity. Why solidarity? Where from such a notion?: In response to this comment we have provided two references [18, 19] for further analysis; however we did not delve further into the connection between solidarity and reciprocity as it would require a lot of space, with the risk of involving concepts and literature far away from the strict focus of the paper.
  3. Lines 246-247: maybe a few words can be added to expand the idea that new risks need the recipients’ activation and responsibility towards social inclusion... The idea is crucial and already stated, but maybe it can be further explained: In response to this comment we explained further the idea regarding the shift from the traditional social risks protection towards the protection from the new social risks (lines 249-252)
  4. Line 314: Ronald Dworkin, not Donald: we corrected the typo
  5. 4: the title and somewhere in the first lines speak about ‘moral foundation and political justification’, but the section does not mention the differences (if any) between the two. The recalling of lucky egalitarianism is surely the path to moral justification: why speak of ‘foundation’ and how to distinguish moral from political justification?: In response to this comment we expanded on the difference and connection between moral foundation and political justification both in the introductory section (lines 68-71 and lines 83-85) and at the beginning of §4 (lines 307-309).

The comments were very useful and helped us a lot to clarify our thoughts.

Thank you so much for your consideration of the revised version of our manuscript.

Sincerely,

Gianluca Busilacchi

Benedetta Giovanola

Reviewer 3 Report

The authors covered in the paper a theoretical and conceptual analysis of the transformations of the European Welfare State. On the basis of deep research studies authors have designed classificatory theoretical framework to better clarify the normative ground, the moral foundation and political justification of the Social Investment Approach and to understand whether it can avoid the risk of erosion of social rights. Due to its broaden comparative and normatively oriented perspective the paper seems to be very important for the contemporary scientific analysis of the Welfare State transformation process. The main hypothesis of the paper is that in order to clarify the normative framework of the SIA, it is necessary to focus on the relationship between social risks and social risks protection and uncover their moral foundation and political justification. The problems raised in the work are deeply embedded in the sciences of society and political systems transformation, but also enter the borderlines of other disciplines: economy, law and social rights. The decisive value of the work is its conceptual character. The strengths of the presented text also include the authors’ choice of the research problem itself. As for the methodology of research, it should be emphasized that it is properly selected. In individual parts of the work, a variety of literature, analysis of documents, articles and content analysis, were skillfully used. I highly appreciate the methodological and systematizing part of the work – presented comparative analysis which leads to the final conceptual approach. The theoretical basis of the text is impressive and the choice of  the literature is significant and adequate for the further discussion run in the paper. In addition, it is necessary to clearly indicate that an important value of the work is the choice of subject matter. The text presented for review is interesting and is certainly characterized by high scientific conceptual approach. The text requires minor editorial and stylistic corrections to strengthen its pragmatism. 

Author Response

(The authors gave the same response as above.)
